# Enhancing Fundamental Movement Competency in Rural Middle School Children Through a Strength Training Intervention: A Feasibility Study

**DOI:** 10.3390/sports13070200

**Published:** 2025-06-22

**Authors:** Janelle M. Goss, Janette M. Watkins, Megan M. Kwaiser, Andrew M. Medellin, Lilian Golzarri-Arroyo, Autumn P. Schigur, James M. Hobson, Vanessa M. Martinez Kercher, Kyle A. Kercher

**Affiliations:** 1Department of Kinesiology, School of Public Health-Bloomington, Indiana University, Bloomington, IN 47405, USAmkwaiser@iu.edu (M.M.K.);; 2Program in Neuroscience, College of Arts and Sciences, Indiana University, Bloomington, IN 47405, USA; 3Department of Epidemiology and Biostatistics, School of Public Health-Bloomington, Indiana University, Bloomington, IN 47405, USA; 4Biostatistics Consulting Center, School of Public Health-Bloomington, Indiana University, Bloomington, IN 47405, USA; lgolzarr@iu.edu; 5White River Valley Middle School, Lyons, IN 47443, USA; 6Department of Health & Wellness Design, School of Public Health-Bloomington, Indiana University, Bloomington, IN 47405, USA; vkercher@iu.edu

**Keywords:** pediatric exercise, physical literacy, sport-based youth development, mobility, endurance

## Abstract

Background: Fundamental movement skills (FMS)—including muscular strength, endurance, and mobility—are linked to better health-related quality of life and higher physical activity in children. Rural children often score lower on FMS tests than urban peers due to resource limitations rather than ability. Thus, increasing access to FMS activities in under-resourced rural areas is essential. The primary objective was to test the feasibility of Hoosier Strength in a rural middle school sample, and the secondary objective was to observe the preliminary changes in FMS-related outcomes pre- to post-intervention and at follow-up. The exploratory objective was to explore how participants responded to different coaches on the Hoosier Strength coaching team (i.e., gender, coaching style during activities). Methods: This study used a Hybrid Type 3 design to evaluate feasibility and FMS outcomes, integrating qualitative and quantitative data. The four-week intervention included a test group (*n* = 24; 14 females, 10 males; mean age: females 12.4 ± 0.5 years, males 12.7 ± 0.4 years) and a control group (*n* = 12; 8 females, 4 males; mean age: females 12.9 ± 0.3 years, males 12.7 ± 0.3 years). Data analysis included descriptive statistics for feasibility indicators (Acceptability of Intervention Measures [AIM], Intervention Appropriateness Measure [IAM], and Feasibility of Intervention Measure [FIM]), linear regression for mobility and muscular endurance changes, *t*-tests for psychological need satisfaction and frustration, and regression analysis for squat knowledge and post-intervention confidence. Results: (1) There was high feasibility across the 4-week Hoosier Strength intervention and at follow-up; (2) there were no statistically significant changes in squat performance; (3) participants’ confidence in their ability to squat at the end of the intervention was significantly predicted by their squat knowledge at baseline; and (4) participants prioritized leadership and team management over tactical analysis, highlighting a preference for coaches who foster teamwork. Conclusions: The findings offer a transparent approach for evaluating the feasibility and preliminary outcomes of the Hoosier Strength intervention in an under-resourced rural middle school, thereby encouraging further investigation into strength training interventions in rural schools.

## 1. Introduction

Fundamental movement skills (FMS) are associated with improved health-related quality of life in children [1]. This competency (measured through muscular strength, muscular endurance, and mobility) is an essential health indicator of school-age children. The literature supports that FMS is best taught during middle school ages as seen in the hourglass model [2]. This model describes how children’s physical activity begins broadly with free play, narrows during early childhood as they develop FMS, and then broadens again as those skills enable participation in a wide range of lifelong physical activities. Likewise, competency in fundamental movements is associated with physical activity levels in childhood and throughout their lifespan [1]. Research supports that rural children tend to have lower performance scores for FMS when compared to their urban counterparts [3]. The observed data discrepancy is not a result of an inability to perform but rather stems from disparities in resources and experiences [4]. Thus, there is a need to improve access to fundamental movement-based physical activity opportunities in rural communities, specifically for children, to promote lifelong health.

Leading organizations in childhood physical activity emphasize the importance of developing FMS during childhood to promote lifelong health and confidence in being active. Project Play, an initiative by the Aspen Institute [5], focuses on increasing youth sports participation through accessible, high-quality programs, offering strategies for creating successful initiatives that support children’s physical activity. These strategies include 8 “plays” (i.e., a framework): (1) ask kids what they want, (2) reintroduce free play, (3) encourage sport sampling, (4) revitalize in-town leagues, (5) think small, (6) design for development, (7) train all coaches, and (8) emphasize prevention. In this study, the intervention, Hoosier Strength, was designed with these “plays” in mind. First, the program is an extension of a published human-centered participatory co-design (a collaborative design approach that actively involves stakeholders in the design process) with children from the target community [6], as well as a multi-level physical activity needs assessment including children [7] (Play 1). Hoosier Strength was also designed to incorporate elements of free play, such as allowing participants to design obstacle courses and challenges while still incorporating new skills (Play 2). Also, the program is an extension of Hoosier Sport, a sport-based youth-development program aiming to expose rural children to a range of sports (Play 3). Moreover, Hoosier Strength was created with development in mind, focusing on foundational FMS such as squats (Play 6). Lastly, the program uses a pilot tested college student-focused implementation strategy [8,9], where college students complete a training course before implementation (Play 7). While none of these characteristics are entirely innovative on their own, since many types of physical activity interventions have been conducted, the combination of them, alignment with Project Play, and emphasis on physical health outcomes creates a relatively novel approach.

Research in physical education (PE) has used self-determination theory [10] to explain motivation to engage in activity. According to self-determination theory’s basic psychological needs mini-theory, autonomous motivation increases when three fundamental psychological needs are fulfilled: autonomy, competence, and relatedness. Autonomy pertains to one’s sense of control during activities. Competence involves perceiving one’s capability to complete tasks effectively. Relatedness encompasses feeling accepted and connected to others. Previous research has indicated that satisfying these basic psychological needs during PE classes correlates with adolescents’ motivation in PE programs [10,11]. Indeed, relatedness among peers and instructors is a predictor for participation in PE, and low feelings of relatedness may result in poor FMS outcomes [11]. Understanding and addressing these fundamental psychological needs within PE not only fosters motivation but also significantly impacts the development of FMS and overall engagement among adolescents.

The present study conducted a four-week, non-randomized controlled feasibility study involving 8 training sessions, with data collection occurring pre-intervention, post-intervention, and a follow-up time point. The study aimed to enhance FMS, specifically focusing on the squat exercise, in 6th and 7th grade children at a rural middle school. The intervention consisted of a progressive squat program implemented by college students within a middle school PE curriculum, gradually increasing exercise difficulty from unloaded to loaded with various axial loadings (e.g., goblet squat, front squat, back squat). The focus of this study was not on resistance training outcomes specifically, but rather feasibility indicators for implementing FMS curriculums into an under-resourced rural middle school. The primary objective was to test the feasibility of Hoosier Strength in a rural middle school sample, and the secondary objective was to observe the preliminary changes in FMS-related outcomes pre- to post-intervention and at follow-up for maintenance. Our primary hypothesis was that the Hoosier Strength intervention would be feasible as defined by “good” scores (median greater than a score of 16) for multiple trial-related (e.g., recruitment capability and retention) and intervention-related feasibility indicators (e.g., Acceptability of Intervention Measures [AIM], Intervention Appropriateness Measure [IAM], and Feasibility of Intervention Measure [FIM]). Our secondary hypothesis was that at the end of our 4-week intervention, the participants would have an increase in FMS outcomes (e.g., range of motion, muscular endurance, load lifted). Our exploratory objective was to explore how participants responded to different coaches on the Hoosier Strength coaching team (i.e., gender, coaching style during activities). This exploratory objective acknowledges the potential impact of coach-participant dynamics on both the feasibility of the intervention and the effectiveness of FMS development. By examining these interactions, we aim to better understand how coach characteristics and behaviors may influence children’s engagement, motivation, and skill acquisition in the context of a rural PE setting.

## 2. Materials and Methods

### 2.1. Setting and Sample

The study sample was comprised of children residing in a rural under-resourced community. The 6th and 7th grade students from a Midwestern middle school participated in the study mainly during PE and health classes. These grade levels were selected because the number of 8th grade students enrolled in PE was too small to form a meaningful sample. Students were selected into the test or control groups as a non-randomized convenience sample based on if they had PE classes this semester or not. Recruitment intervention, assessments, and evaluations occurred in the school setting, which offered a practical and well-structured environment. Figure 1 provides further detail of the conceptual framework of Hoosier Strength which focuses on a multilevel approach and multiple intervention strategies, and emphasizes basic psychological needs when targeting outcomes. Thirty-five participants were included, with 24 in the test group (underwent the intervention) and 12 in the control group (had no PE class and therefore did not receive intervention). See Table 1 for more detail for each group.

### 2.2. Design

The present study used a non-randomized controlled Hybrid Type 3 design, which allows for simultaneously assessing implementation feasibility indicators while evaluating clinical outcomes [12]. A Hybrid Type 3 design offers a framework that integrates qualitative and quantitative data, enabling a formative evaluation of both the intervention’s preliminary signals and assessment of its implementation feasibility [12]. The intervention comprised two in-person strength training sessions per week over four weeks (8 total sessions), along with one weekly remote homeroom activity session.

Prior to the intervention, all research assistants underwent an online video and quiz assessment to ensure their competency in the movements and skills being taught. At the start of the intervention (week 1), the research team collected squat-based measurements (e.g., squat range of motion was measured with a goniometer, total squats performed in 1 min) to establish a baseline for performance. Baseline data collection took place on Tuesday of week 1 and lasted approximately 20 min per student. After completing their assessments, students participated in 24 min of exposure to the squat equipment and movement-based games designed to familiarize them with activities included in the intervention. The intervention officially began on Thursday of week 1 and was conducted for the full 44 min session. In week 4, the final intervention session was held on Tuesday, and post-intervention data collection occurred on Thursday. At the post-intervention data collection, a survey was given to gauge psychological outcomes (see Section 2.4.2) and a follow-up assessment was conducted which included repeating the squat measurements. Feasibility data was collected over 4 weeks (e.g., attendance and retention). After completing their assessments, students were permitted to engage in free play in the gym for the remainder of the 44 min period. In total, students participated in six full intervention sessions over the course of the four-week program.

Following the completion of the 4-week intervention, Hoosier Strength was completed (the intervention that is the focus of this paper) and another physical activity intervention began through the Hoosier Sport Lab (the overseeing lab that Hoosier Strength is based in). After the Hoosier Sport Lab finished the semester, the Hoosier Strength team returned to the middle school for a 4-week post-intervention follow-up to assess retainment of squat competency (this was at the 8-week mark from week 1 of Hoosier Strength). See Figure 2 for an overview of the intervention’s key time points, which include a 4-week strength training-focused intervention, a subsequent program by the lab which Hoosier Strength existed within, and data collection.

### 2.3. Intervention

In-person strength activities, lasting around 44 min for each grade-based group, took place twice per week. These sessions were conducted by research assistants. Likewise, once per week research assistants conducted one remote session via Zoom to promote physical activity within the classroom setting. These activities also included educational components on physiology related to squats. This systematic data collection approach was designed to provide insights into the participants’ experiences, changes in physical activity behaviors, and increases in squat competency during the study.

### 2.4. Measures

#### 2.4.1. Squat Competency

The study assessed lower extremity mobility, muscular strength, and muscular endurance. Mobility was measured at week 1, week 4, and follow-up (week 8) in terms of squat depth, with a goal of 90 degrees (e.g., parallel). Squat depth was assessed with a pitch and slope locator. The participants’ heels had to stay on the ground during the measurement with the feet approximately shoulder width apart. Strength was measured at week 4 and follow-up based on the maximum load that participants could squat to a box that was at parallel height. Muscular endurance was measured at week 1, week 4, and at follow-up via a 60 s squat test to a box at 90 degrees (higher number of repetitions equaled higher muscular endurance). Strength was not evaluated in the baseline data collection because participants lacked the necessary experience to safely perform loaded squat movements.

#### 2.4.2. Psychological Outcomes

Psychological needs were assessed with the Basic Psychological Needs Satisfaction and Frustration Scale (BPNSFS). The BPNSFS is a self-reported scale related to self-determination theory that assesses autonomy, competence, and relatedness. In a validation study of the original BPNSFS conducted in four countries, internal consistency values ranged from 0.64 to 0.89, with subscale reliability in the US varying from 0.71 to 0.89 [13]. The validity of the scale was also supported in adolescents [14]. For our analyses, we computed an overall score.

#### 2.4.3. Trial-Related Feasibility Indicators

Recruitment capability and retention were measured in terms of attendance during in-person and homeroom activities.

#### 2.4.4. Intervention-Related Feasibility Indicators

The study assessed the feasibility of the intervention through various indicators, including attendance, acceptability, appropriateness, and overall feasibility. Adapted AIM, IAM, FIM scales (adapted to be appropriate for children) were used at follow-up. These original scales were validated to assess feasibility metrics [15], but the adapted scales were not validated. Each measure has shown satisfactory psychometric properties in adult populations, encompassing content validity, reliability, structural validity, structural invariance, known-groups validity, and responsiveness to change [15]. A Likert scale ranging from “completely disagree” to “completely agree” was utilized to gauge responses for feasibility scales. Summary scores for each measure were calculated using means/medians and standard deviations. Additionally, participant attendance was evaluated to measure compliance.

### 2.5. Procedure

Children from a rural middle school were recruited through flyers distributed during school lunch hours and emails sent to parents. The parents of interested children were contacted via email, and a parental call was scheduled to explain the study and obtain verbal consent. Additionally, parents completed the Physical Activity Readiness Questionnaire (PAR-Q) to assess their child’s eligibility and provide written consent via an online survey [16]. Children provided their assent during week 1 through a survey provided in person with the research team, where they could choose to participate or not. Research assistants led classes where 6th and 7th grade students attended in their respective time blocks. Each session lasted 44 min and included a dynamic warm-up, skill development, child-friendly drills, games that incorporated key movement patterns (e.g., obstacle courses with squats), and a brief cool-down activity (e.g., stretching the muscle groups targeted that day). The intervention followed a periodized progression across the four weeks. In week 1, students performed bodyweight movements. In week 2, they progressed to front-loaded squats, such as goblet squats using weights ranging from 5 to 35 pounds. Week 3 introduced axially loaded front squats using dowel bars ranging from 5 to 30 pounds. In week 4, students performed axially loaded back squats with similar dowel bar weights (5 to 30 pounds). For pre-and post-intervention data collection, children would leave their regularly scheduled class and go to the gym to complete the assessments. If children were absent on a data collection day, a research assistant would go back on another day to collect their data.

### 2.6. Data Analysis

Data was analyzed using R Studio (2023.12.0 + 369). Descriptive statistics were used to analyze intervention-related feasibility indicators, including the AIM, the IAM, and the FIM. A linear mixed model was conducted to assess differences in mobility (e.g., range of motion) and muscular endurance (e.g., 60-s squat test) at week 1, week 4, and at follow-up (week 8). T-tests evaluated changes in psychological need satisfaction and frustration pre- and post-intervention. Sample size limitations may limit statistical interpretations, but that is warranted as this is a pilot/feasibility study and differences from potential clinical implications will be addressed. Additionally, regression analysis examined whether baseline squat knowledge predicted post-intervention squat confidence.

## 3. Results

### 3.1. Objective 1

The primary objective was to test the feasibility of Hoosier Strength in a middle school sample. Using basic descriptive statistics, we calculated the median, mean, and SD for scores of intervention-related feasibility indicators (AIM, IAM, FIM). Table 2 shows these results.

For trial-related feasibility indicators, recruitment exceeded the goal of 20 participants for the pilot study (*n* = 36). Moreover, there was a retention of over 97%, with only one participant dropping out of the program (due to moving schools).

### 3.2. Objective 2

Our secondary hypothesis was that at the end of our 4-week intervention, the participants would have an increase in mobility (e.g., squat depth) and muscular endurance (e.g., 60 s squat test), and a positive trend in BPNSFS and psychological (e.g., confidence) outcomes. We conducted a linear mixed model to assess if there were differences in the means at week 1 (i.e., pre), week 4 (i.e., post), and week 8 (i.e., follow-up). For ROM, the test group showed no significant changes from pre (80.05 ± 14.41) to post (77.86 ± 17.83), or pre to follow-up (79.77 ± 14.76). Similarly, the control group did not show significant changes from pre (79.17 ± 13.11) to post (84.64 ± 11.53) or pre to follow-up (79.09 ± 14.97). As shown in Table 3, there were no significant effects of group, time, or group-by-time interaction. The effect sizes for ROM are reported in Table 4.

Additionally, muscular endurance was analyzed using a linear mixed model (Table 5). The analysis for the test group showed a non-significant result from pre (37.18 ± 10.31) to post (41.05 ± 8.92), and pre to follow-up (42.09 ± 11.14). Similarly, the control group did not show significant changes from pre (35.42 ± 7.35), to post (44.36 ± 7.95), or pre to follow-up (41.64 ± 8.73). As shown in Figure 3 and Table 5, there were no significant effects of group, time, or group-by-time interaction. The effect sizes for muscular endurance are reported in Table 6 and the difference-in-difference results are shown in Table 7.

The BPNSFS was used to assess changes in psychological need satisfaction from pre- to post-intervention. In the test group, the mean psychological need satisfaction decreased slightly from 36.5 (*SD* = 5.94) pre-intervention to 34.4 (*SD* = 7.77) post-intervention. However, this change was not statistically significant (*t*(22) = 1.002, *p* = 0.322). In the control group, the mean psychological need satisfaction increased marginally from 35.9 (*SD* = 7.06) pre-intervention to 36.4 (*SD* = 4.61) post-intervention, but this difference was also not statistically significant (*t*(10) = −0.178, *p* = 0.860).

Moreover, psychological need frustration was explored. In the test group, the mean frustration score decreased from 36.5 (*SD* = 5.94) pre-intervention to 20.36 (*SD* = 7.43) post-intervention. However, this decrease was not statistically significant (*t*(22) = −0.445, *p* = 0.658). The control group’s frustration score also decreased, from 20.09 (*SD* = 9.181) pre-intervention to 16.90 (*SD* = 7.40) post-intervention, but this difference was not significant (*t*(10) = −0.894, *p* = 0.382).

Additionally, a regression was performed to determine if confidence to squat post-intervention was predicted by squat knowledge pre-intervention for the entire sample. The regression was significant (*F*(14,17) = 2.337, *p* = 0.049), indicating that baseline squat experience accounted for 37.65% (adjusted R^2^ = 0.3765) of the variance in squat confidence outcomes. Squat confidence for the test group decreased slightly from pre-intervention (*m* = 0.82, *SD* = 0.38) to post-intervention (*m* = 0.77, *SD* = 0.42). In the control group, squat confidence also decreased from pre-intervention (*m* = 0.81, *SD* = 0.40) to post-intervention (*m* = 0.45, *SD* = 0.52). Figure 4 shows these changes in squat confidence from pre- to post-intervention for both test and control groups. Although both groups experienced a decrease in confidence, these changes were not statistically significant (*p* > 0.05).

Finally, the relationship between the BPNSFS score at baseline and squat outcomes post-intervention was examined. The BPNSFS score was not a significant predictor of squat confidence (*F*(7,24) = 0.861, *p* = 0.550), range of motion (*F*(1,31) = 1.044, *p* = 0.314), or repetitions (*F*(1,31) = 0.860, *p* = 0.360). Figure 5 illustrates that BPNSFS scores were not significant predictors of post-intervention squat outcomes, including confidence, range of motion, or repetitions (all *p* > 0.05), indicating that psychological need satisfaction at baseline was not significantly related to the measured squat outcomes.

### 3.3. Objective 3: Exploratory Data

In addition to measuring squat metrics, participant preferences for coaching styles were also measured via a survey in week 4. Seventeen participants indicated that during past experiences they most enjoyed being instructed by male coaches, while 16 participants indicated female coaches. However, 60% of participants indicated no preference for a coach’s gender. Participants prioritized leadership and team management over tactical analysis, indicating a preference for coaches who foster teamwork. This data is shown in Figure 6.

## 4. Discussion

The present study had three objectives: to test the feasibility of Hoosier Strength; to identify potential changes in squat performance (e.g., range of motion and muscular endurance) and psychological outcomes (e.g., fulfillment of basic psychological needs, confidence to squat); and to explore the coaching preferences of middle school students. The study yielded four key findings: (1) the trial- and intervention-related feasibility indicators were consistently high throughout the 4-week intervention and at follow-up, indicating that the Hoosier Strength program is feasible to implement; (2) there were no significant changes in squat performance outcomes over the 4-week intervention; (3) participants’ confidence in their ability to squat at the end of the intervention was significantly predicted by their squat knowledge at baseline; and (4) participants prioritized leadership and team management over tactical analysis, indicating a preference for coaches who foster teamwork. These findings offer a transparent approach for evaluating the feasibility and preliminary outcomes of the Hoosier Strength intervention, thereby encouraging further investigation into strength training interventions in rural schools.

For our first key finding, Hoosier Strength was found to have good feasibility for the targeted population (Table 2). This was an important measurement based on pre-existing literature. For example, similar research has found that feasibility scores (AIM, FIM, IAM) pre-intervention correlated with the success of an intervention focused on youth physical activity levels [17]. Additionally, our utilization of FIM, IAM, and AIM scales is supported by other studies as a validated method [15,18]. Both AIM and IAM assess aspects of intervention readiness; however, AIM focuses on acceptability, measuring stakeholders’ attitudes and reactions toward the intervention, while IAM focuses on appropriateness, evaluating the fit and relevance of the intervention to the specific context or setting. Likewise, FIM assesses the feasibility of implementing the intervention within a specific context or setting, such as resource availability, organizational support, and logistical constraints. Our previous pilot work has found that Hoosier Sport programs report high feasibility, acceptability, and appropriateness as reported by the children [6]. This pilot study specifically found that intervention-related scores (FIM, AIM, and IAM) consistently surpassed the “good” threshold, and the program had a 100% retention and recruitment success.

Furthermore, children’s opinions are rarely incorporated into the research process [19]. An important component of Hoosier Strength, in line with Project Play’s Play 1 (i.e., ask kids what they want), is assessing the participants’ opinions of the intervention. This is achieved specifically in week 4 of the intervention by utilizing the feasibility scales (FIM, IAM, AIM) in a survey taken by the participants (middle school-aged children). This allows for Hoosier Strength to evolve with the wants and needs of the children in mind. Overall, participants reported a positive perception of the intervention. The data showed high scores for accessibility, appropriateness, and feasibility (scored above the median of 16), indicating positive participant perception of Hoosier Strength (Table 1). For future applications, studies should consider utilizing these feasibility indicators, specifically targeted at youth participants.

Second, physiological performance changes were not identified following Hoosier Strength. Squat ROM and muscular endurance showed no significant difference pre- and post-intervention, implying limited statistical impact on this aspect of squat performance (see limitations section for environmental factors). Although non-significant, post-intervention muscular endurance gains (Control: +25%, Test: +10%) suggest there may be clinical relevance for future longer-duration trials. Other research has found that school-based strength interventions can increase resistance training-related repetitions [20] and ROM [21]. One potential reason for our non-significant results is that focusing on squat technique with lower repetitions, rather than higher repetitions/volume, may have influenced the outcomes. Furthermore, four-week interventions may be insufficient for neuromuscular adaptations; 8–12 weeks are typical for significant FMS gains [22]. Our low-volume programming prioritized technique over load, potentially limiting endurance gains vs. higher-repetition protocols [23]. Research suggests that training with higher reps under load can improve muscular endurance in youths [20]. The decision to use this lower-volume approach in our intervention was based on a focus on physical literacy and working within the confines of an existing program. Additionally, the non-significant changes in squat ROM could also be due to lack of duration and frequency of the intervention. Other research has shown that resistance training can improve ROM, though no optimal duration has been established [21]. Future studies should investigate the ideal duration and frequency of resistance training to enhance ROM. Additionally, our sample participated in Hoosier Sport, a physical activity program focused on sports other than strength training (see Section 5: Limitations), which may have elevated their baseline ROM compared to untrained adolescents. This may have limited the amount of change observed. Nonetheless, improving ROM remains essential for enhancing physical literacy. For example, research has found that range of motion supports FMS (e.g., running, jumping, squatting) through more efficient and effective movement patterns [22]. This aligns with Project Play’s Play 6 (i.e., design for development). Our future goal is to continue to expand on this program to allow for development across all components of FMS competency.

The third finding concerns how baseline knowledge might predict psychological outcomes, specifically confidence in squatting, post-intervention. This finding revealed that participants’ knowledge of the squat movement at the start of the intervention significantly predicted their confidence in performing squats at the end. This suggests that enhancing initial knowledge of exercises could be a valuable strategy for boosting confidence and improving overall performance in strength training programs. A recent scoping review supports the overall concept of physical literacy/knowledge promoting performance and self-efficacy but results remain heterogeneous and more research is needed [24]. Indeed, some programs may benefit from initially increasing knowledge of movements or exercise (i.e., physical literacy) before progressing to teaching specific skills and movements. This approach is supported by other research suggesting that promoting physical literacy can aid in the development of fundamental movement skills [25]. Additionally, research supports that an increase in autonomy in exercise or in terms of motivation is predictive of exercise participation and the extent to which one will participate [26]. Based on our findings and current literature, we believe that future research should focus on establishing a strong baseline knowledge of physical literacy before implementing physical practice, as this could enhance intervention outcomes.

Finally, we explored the coaching preferences of middle school students. Based on survey questions, we assessed certain preferences the students had for various coaching traits. A significant number of participants expressed no preference for having either a male or a female coach in Hoosier Strength, but the past experiences of the participants showed preferences for the gender of coaches (e.g., which gender they preferred as a coach in experiences outside of Hoosier Strength, with 50% preferring male coaches). Other research substantiates our findings of no preference for a coach’s gender [27]. However, empirical work exploring gender preferences in coaching has found that youth sports are typically dominated by male coaches, which could hinder a safe, positive, and developmentally supportive sports experience based on characteristics male coaches tend to possess [28]. Combined, these studies highlight the importance of diverse coaching options. Also, the participants reported leadership and team management as the most important coaching value, while game preparation and analysis ranked lowest. This suggests that the targeted population prefers an environment that fosters teamwork rather than skill work alone. Team-focused approaches may be a valuable strategy for designing future interventions in rural contexts. Also, in line with Project Play’s Play 7 *(Train all coaches*)*,* all coaches of Hoosier Strength underwent training before leading the participants through the program. This effort to ensure competent coaches was reflected by the findings that there was not one coach who was preferred over another. These findings support the use of college-aged students to coach/implement FMS-related interventions in youth populations.

In conclusion, Hoosier Strength demonstrated high feasibility in a rural school. While physiological outcomes were statistically unchanged post-intervention, baseline knowledge predicted confidence gains. Future studies should extend the duration, increase sample sizes, and prioritize coaching strategies that foster teamwork.

## 5. Limitations

This study had several limitations. First, the sample size was not randomized, which affects the ability to interpret causal relationships. This lack of randomization was justified based on the feasibility of working within the school’s existing class structure. We plan to add randomization to future larger-scale study designs with multiple schools. Second, the sample size was relatively small, with only 36 participants. The control group consisted of 12 participants, while the test group had 24 participants. The smaller size of the control group led to greater variability in their data, complicating the comparison between the test and control groups. However, a strength of this study was the pragmatic design, which focused on practical and applicable outcomes, increasing the likelihood that the findings are relevant to real-world settings. Third, the control group had prior exposure to physical activity education and training through Hoosier Sport, another intervention program aimed at promoting physical activity. This prior experience could have enhanced their performance on the FMS competency tests, potentially biasing the results. Also, the strength assessment was limited by the maximum load used in the study. The heaviest weight was a 30-pound dowel bar, which all participants could successfully squat to a box at parallel. To accurately measure changes in muscular strength over the intervention period, a broader range of weights is necessary. Also, during the squat endurance test, students squatted to a box set at a height parallel to their knees. This provided an external cue for squat depth, which may have allowed students of varying ability levels to perform high repetitions. A loaded squat could have better distinguished students with higher FMS competency by requiring more muscle recruitment and making depth more challenging compared to an unloaded squat. Additionally, future studies should consider increasing the duration of the intervention to allow sufficient time for physiological changes to occur. Lastly, while our data showed that both satisfaction and frustration scores declined, this outcome was unexpected, as these two variables typically work in opposition to one another. The lack of statistical significance in both outcomes may suggest that the intervention’s effects were too subtle to be detected by the BPNSFS within the study’s timeframe or sample size. This implies that while the intervention may have had some influence, its impact on psychological need satisfaction and frustration might not have been strong enough to produce measurable changes within the study’s design.

## Figures and Tables

**Figure 1 sports-13-00200-f001:**
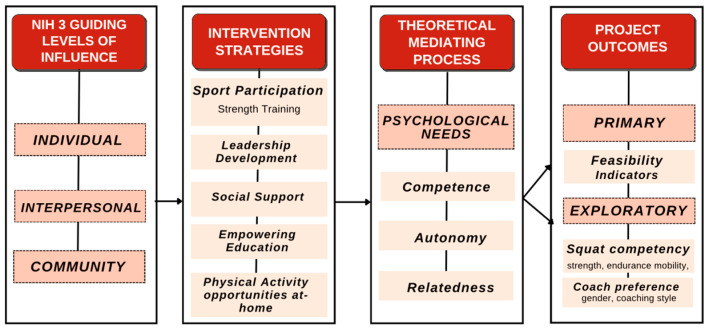
Hoosier Strength conceptual framework.

**Figure 2 sports-13-00200-f002:**
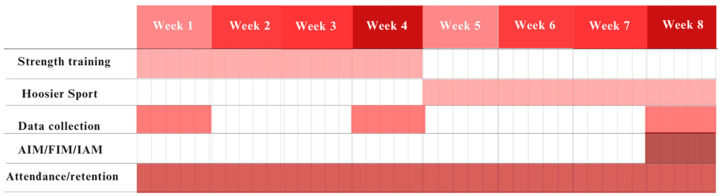
Timeline of data collection.

**Figure 3 sports-13-00200-f003:**
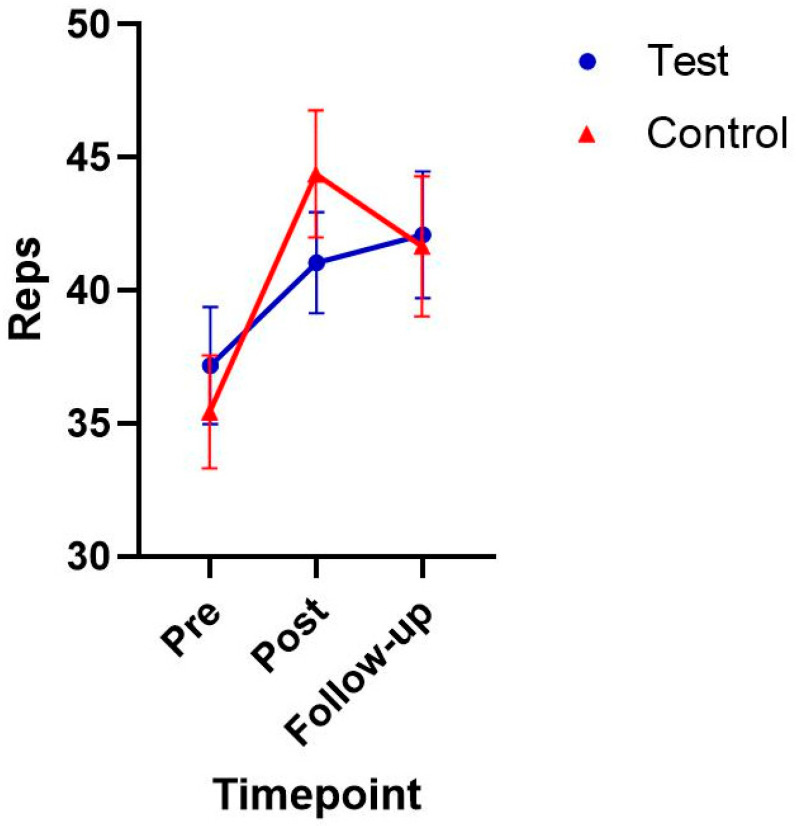
Muscular endurance (reps) in test and control groups.

**Figure 4 sports-13-00200-f004:**
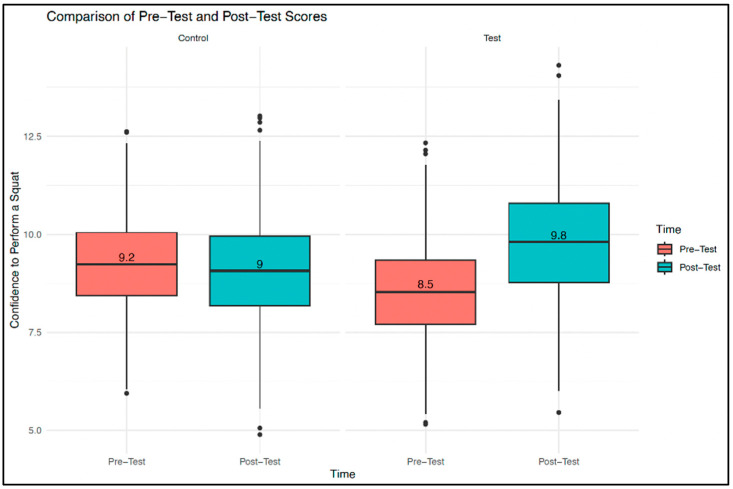
Participant confidence to squat.

**Figure 5 sports-13-00200-f005:**
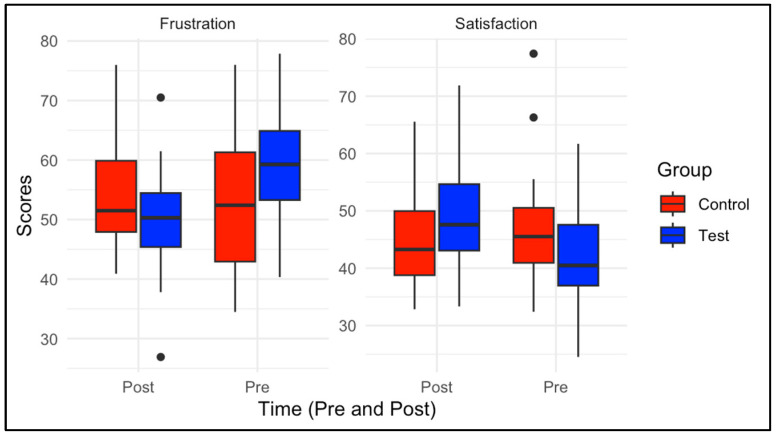
Frustration and satisfaction pre- to post-intervention for test and control groups.

**Figure 6 sports-13-00200-f006:**
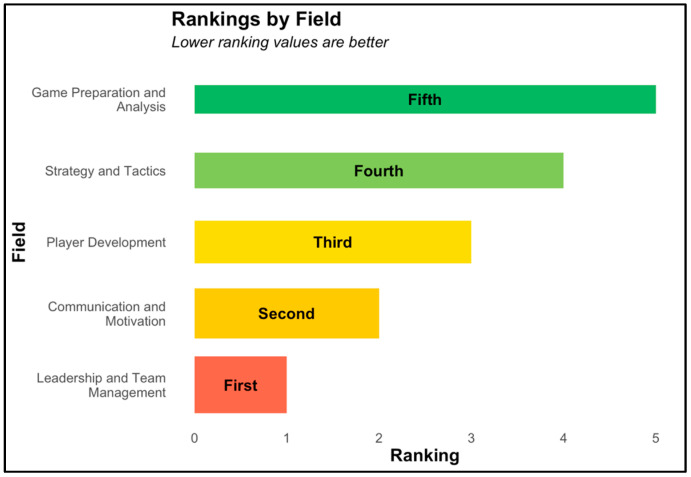
Participant preference for coaching values.

**Table 1 sports-13-00200-t001:** Demographics.

	Test Group	Control Group
**Gender (n)**		
Female	14	5
Male	10	6
**Age (M, SD)**		
Female	12.4 (0.5)	12.9 (0.3)
Male	12.7 (0.4)	12.7 (0.3)

**Table 2 sports-13-00200-t002:** Feasibility data for Hoosier Strength.

Group	Mean	Median	SD
FIM			
Control	17.27	18	2.10
Test	16.36	16	3.38
IAM			
Control	17.54	18	2.62
Test	16.09	17	4.87
AIM			
Control	18.54	19	1.43
Test	16.72	16.5	3.38

**Table 3 sports-13-00200-t003:** Linear mixed model results for range of motion.

Characteristic	Beta	95% CI	*p*-Value
Group			
Control	–	–	–
Test	1.1	−9.7, 12.0	0.836
Time			
Pre	–	–	–
Post	4.4	−3.5, 12.0	0.270
Follow-up	−1.2	−9.0, 6.7	0.771
Group*Time			
Test*Post	−7.4	−17, 2.4	0.135
Test*Follow-up	0.06	−9.7, 9.8	0.990

Note: CI = confidence interval. * = interaction.

**Table 4 sports-13-00200-t004:** Effect sizes for range of motion.

**Time**	**Group**	**ES**	**SE**	**df**	**Lower CL**	**Upper CL**
Pre-Post	Control	−0.471	0.425	55.082	−1.322	0.381
Pre-Follow-up	Control	0.124	0.423	55.082	−0.725	0.973
Post-Follow-up	Control	0.594	0.429	58.667	−0.263	1.452
Pre-Post	Test	0.322	0.310	58.662	−0.298	0.943
Pre-Follow-up	Test	0.118	0.309	58.662	−0.501	0.737
Post-Follow-up	Test	−0.205	0.302	58.662	−0.809	0.400
**Time**	**Contrast**	**ES**	**SE**	**df**	**Lower CL**	**Upper CL**
Pre	Control-Test	−0.120	0.578	55.082	−1.279	1.038
Post	Control-Test	0.672	0.592	58.662	−0.512	1.857
Follow-Up	Control-Test	−0.127	0.590	58.662	−1.307	1.054

Note: ES = effect size; Cohen’s *dz* interpretation: small = 0.2, medium = 0.5, large = 0.8; SE = standard error; CL=confidence limits.

**Table 5 sports-13-00200-t005:** Linear mixed model results for muscular endurance.

Characteristic	Beta	95% CI	*p*-Value
Group			
Control	–	–	–
Test	1.5	−5.3, 8.2	0.666
Time			
Pre	–	–	–
Post	9.3	−3.4, 15.0	0.002
Follow-up	6.5	0.69, 12.0	0.029
Group*Time			
Test*Post	−5.2	−12, 2.1	0.135
Test*Follow-up	−1.4	−8.6, 5.8	0.699

Note: CI = confidence interval. * = interaction.

**Table 6 sports-13-00200-t006:** Effect sizes for muscular endurance.

**Time**	**Group**	**ES**	**SE**	**df**	**Lower CL**	**Upper CL**
Pre-Post	Control	−1.334	0.433	67.344	−2.198	−0.469
Pre-(Follow-up)	Control	−0.941	0.428	67.344	−1.795	−0.088
Post-(Follow-up)	Control	0.393	0.427	70.878	−0.460	1.245
Pre-Post	Test	−0.588	0.311	70.864	−1.207	0.031
Pre-(Follow-up)	Test	−0.739	0.312	70.864	−1.361	−0.116
Post-(Follow-up)	Test	−0.150	0.302	70.864	−0.752	0.451
**Contrast**	**Time**	**ES**	**SE**	**df**	**Lower CL**	**Upper CL**
Control-Test	Pre	−0.211	0.487	67.344	−1.183	0.761
Control-Test	Post	0.535	0.501	70.864	−0.464	1.533
Control-Test	Follow-up	−0.008	0.499	70.864	−1.003	0.987

Note: ES = effect size; Cohen’s *dz* interpretation: small = 0.2, medium = 0.5, large = 0.8; SE = standard error; CL = confidence limits.

**Table 7 sports-13-00200-t007:** Difference-in-difference results for muscular endurance.

Group	Time	EST	SE	df	*t* ratio	*p*-Value
Test-Control	Post-Pre	−0.211	3.63	62.67	−1.43	0.158
Test-Control	(Follow-up)-Pre	0.535	3.63	62.67	−0.39	0.699
Test-Control	(Follow-up)-Post	−0.008	3.63	60.40	1.04	0.303

Note: SE = standard error; EST = estimate.

## Data Availability

The data presented in this study are available on request from the corresponding author due to participant confidentiality and privacy considerations.

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
