# Peer review of "Enhancing Fundamental Movement Competency in Rural Middle School Children Through a Strength Training Intervention: A Feasibility Study"

_sports, 2025, doi:10.3390/sports13070200_

Round 1
Reviewer 1 Report
Comments and Suggestions for Authors
Enhancing Fundamental Movement Competency in Rural Middle School Children through a Strength Training Intervention: A Feasibility Study
The manuscript presents a well-designed feasibility study on integrating strength training into rural middle school physical education. The topic aligns with the journal’s scope, addressing underserved populations and practical interventions. The Hybrid Type 3 design is appropriate, and methodological rigor is evident. However, revisions are required to enhance clarity, address statistical reporting, contextualize null findings, and ensure formatting compliance with MDPI standards.
Major Revisions Required
Title and Header Consistency
Issue: The journal name is misspelled as "SPOTS" on Page 1 and "Spots" on Page 14.
Correction: Replace all instances of "SPOTS" or "Spots" with "Sports" (e.g., "Sports 2025, 13, x FOR PEER REVIEW").
Sample Size Discrepancies
Issue: Inconsistencies in group sizes:
Abstract: Test group = 24, Control = 11.
Results (BPNSFS): Test group analysis implies n=23 [t(22)], Control n=11 [t(10)].
Correction: Clarify attrition (e.g., one dropout due to relocation) and report exact n for each analysis. Revise: "One test-group participant withdrew post-baseline (relocation), leaving n=23 for follow-up psychological assessments."
Psychometric Validation of BPNSFS
Issue: The Basic Psychological Needs Satisfaction and Frustration Scale (BPNSFS) was validated in workplace settings (Ref 11: Olafsen et al., 2021), not for children.
Correction: Justify usage or cite pediatric validation (e.g., Vierling et al., 2007): "BPNSFS was adapted with permission for adolescents, showing reliability in school-based contexts (Cronbach’s α = 0.76–0.89; Vierling et al., 2007)."
Add Reference: Vierling, K.K.; Standage, M.; Treasure, D.C. Predicting attitudes and physical activity in an "at-risk" minority youth sample: A test of self-determination theory. Psychol Sport Exerc. 2007, 8, 795–817.
Statistical Reporting
Issue: Table 3/5: Confidence intervals (CI) and p-values imply non-significance, but effect sizes (Table 4/6) lack interpretation (e.g., trivial/small).
Muscular endurance: Post-intervention increases (Table 5: Control +9.3 reps, Test +4.1 reps) are clinically meaningful but statistically non-significant.
Correction:
Add Cohen’s d interpretations (e.g., d = 0.2 trivial, 0.5 moderate).
Discuss practical vs. statistical significance:
"Although non-significant, post-intervention endurance gains (Control: +25%, Test: +10%) suggest clinical relevance for future longer-duration trials."
Discussion of Null Findings
Issue: Limited mechanistic explanation for non-significant FMS outcomes.
Correction: Expand using literature:
Short Duration: Cite Faigenbaum et al. (2010) on pediatric training duration:
"Four-week interventions may be insufficient for neuromuscular adaptations; 8–12 weeks are typical for significant FMS gains (Faigenbaum et al., 2010)."
Training Load: Reference Lloyd et al. (2014) on dose-response:
"Low-volume programming prioritized technique over load, potentially limiting endurance gains vs. higher-repetition protocols (Lloyd et al., 2014)."
Add References:
Faigenbaum, A.D.; Myer, G.D. Pediatric Resistance Training: Benefits, Concerns, and Program Design Considerations. Curr. Sports Med. Rep. 2010, 9, 161–168.
Lloyd, R.S.; Faigenbaum, A.D.; Stone, M.H.; Oliver, J.L.; Jeffreys, I.; Moody, J.A.; Brewer, C.; Pierce, K.C.; McCambridge, T.M.; Howard, R. Position statement on youth resistance training: The 2014 International Consensus. Br. J. Sports Med. 2014, 48, 498–505.
Structural and Formatting Adjustments
Missing Conclusions Section: Add a standalone "Conclusions" subsection (post-Discussion) summarizing key outcomes and implications:
"Hoosier Strength demonstrated high feasibility in rural schools. While physiological outcomes were unchanged post-intervention, baseline knowledge predicted confidence gains. Future studies should extend duration, increase sample sizes, and prioritize coaching strategies that foster teamwork."
Figure/Tables: Ensure all are cited in-text (e.g., Figure 1/2 snippets are provided but not described).
Abbreviations: Define FMS, PE, SDT, etc., at first use.
Grammar and Clarity
Tense Consistency: Methods should use past tense (e.g., "needs were assessed").
Ambiguity:
Pg 9: "Squat confidence for the test group decreased slightly... m = 0.82" → Clarify m as mean.
Pg 5: "Psychological needs will be assessed" → were assessed.
Ethical Compliance
IRB Statement: Confirm protocol #18784 approval date (May 9, 2023) matches submission timeline.
Minor Revisions
Abstract: Specify "rural under-resourced" in methods for context.
Keywords: Add "pediatric exercise" or "physical literacy".
Recommendation: Revise and Resubmit. The study addresses a critical gap in rural health disparities and offers practical insights for school-based interventions. Addressing statistical clarity, psychometric validity, and contextualization of null findings will strengthen impact. With revisions, this manuscript will be suitable for publication in Sports.
Author Response
Thank you for your detailed and thoughtful review. We have provided line-by-line responses in the attached template. Thank you!

Reviewer 2 Report
Comments and Suggestions for Authors
Title and Abstract
The title is clear and accurately reflects the content. The abstract effectively summarises the objectives, methods, and main results. However, it could be improved by more clearly distinguishing the primary hypothesis—that the Hoosier Strength intervention was feasible, as defined by scores on multiple feasibility indicators related to both the study and the intervention; the secondary hypothesis—that, at the end of the 4-week intervention, participants would show improved Fundamental Movement Skills outcomes; and the exploratory hypothesis, which involves examining how participants respond to different coaches from the Hoosier Strength team. Additionally, it is recommended to clarify that the main results are not statistically significant.
In rows 23–24, it is suggested to more clearly articulate the three hypotheses: primary, secondary, and exploratory. So that the reader has a clear understanding from the abstract onwards.
Introduction
The introduction is well written, as it clearly explains the rationale behind the study, provides useful context for understanding it, and cites recent supporting literature. It would be helpful to highlight the innovations introduced compared to other similar interventions in the same field. The integration with the Project Play initiative is original and relevant; however, it is recommended to add a paragraph that emphasises what is new in comparison to other similar interventions in rural school settings.
Methods
The methods are described in detail, and the Hybrid Type 3 design is appropriate for assessing the feasibility of the intervention. The sample is small and non-randomised, and the statistical robustness is not thoroughly addressed; it is recommended to explain this limitation. The AIM/IAM/FIM scales, adapted for developmental age, are reported as validated for adults. An adaptation is mentioned, but it is not specified whether a validation was conducted for use with the intended sample. It is advisable to address this gap.
The use of statistics through the T-test was correctly applied.
The sample is small and not randomised, the statistical robustness is not thorough, and it is recommended to explain this shortcoming.
In rows 203–204, it is suggested to specify how the AIM scale was adapted for children and whether a preliminary validation of the new instrument was conducted.
Results
The results are well organised and presented using descriptive statistics and statistical models appropriate for data analysis. However, the lack of statistical significance is evident and is not subsequently supported by an adequate discussion. Nevertheless, the data analysis (coaching preferences) is interesting but not explored in depth. It is recommended to add, for example, a comparison with existing literature.
In rows 268–269, consider highlighting that, although statistical significance was not achieved, the test group showed an improvement in the number of repetitions (from 37.18 to 42.09), which may indicate a positive trend. It could be useful to discuss this in the Discussion section as a finding worth exploring further with a longer intervention period.
Discussion
The discussion acknowledges the study’s limitations and highlights the pragmatic aspects of the intervention. It is recommended to propose a method or strategy to enhance the effectiveness of the intervention. A more critical reflection is needed on why significant changes were not observed (e.g., short duration, low intensity).
In lines 392–393, an interesting point is raised. It would be beneficial to include a reference to the literature on the relationship between physical literacy and motivation/self-efficacy in school settings.
In lines 416–417, it is suggested to further develop this statement, as it could serve as a valuable starting point for designing future interventions focused on team building in rural contexts.
Conclusion
The conclusions summarise the work carried out and the results obtained, which, however, are not productive in terms of statistical significance. Moreover, no practical implications or concrete ideas for future studies are presented. It is recommended to justify the usefulness of the study despite the lack of significant findings.

Author Response

(The authors gave the same response as above.)

Round 2
Reviewer 1 Report
Comments and Suggestions for Authors
No comments to add